# Intrinsic Point Defects in Silica for Fiber Optics Applications

**DOI:** 10.3390/ma14247682

**Published:** 2021-12-13

**Authors:** Giuseppe Mattia Lo Piccolo, Marco Cannas, Simonpietro Agnello

**Affiliations:** 1Dipartimento di Fisica e Chimica, Università Degli Studi di Palermo, Via Archirafi 36, 90123 Palermo, Italy; simonpietro.agnello@unipa.it; 2Dipartimento di Fisica e Astronomia, Università Degli Studi di Catania, Via Santa Sofia 64, 95123 Catania, Italy; 3ATeN Center, Università Degli Studi di Palermo, Viale delle Scienze Ed. 18, 90128 Palermo, Italy

**Keywords:** silica point defects, optical fibers, radiation effects

## Abstract

Due to its unique properties, amorphous silicon dioxide (*a*-SiO_2_) or silica is a key material in many technological fields, such as high-power laser systems, telecommunications, and fiber optics. In recent years, major efforts have been made in the development of highly transparent glasses, able to resist ionizing and non-ionizing radiation. However the widespread application of many silica-based technologies, particularly silica optical fibers, is still limited by the radiation-induced formation of point defects, which decrease their durability and transmission efficiency. Although this aspect has been widely investigated, the optical properties of certain defects and the correlation between their formation dynamics and the structure of the pristine glass remains an open issue. For this reason, it is of paramount importance to gain a deeper understanding of the structure–reactivity relationship in *a*-SiO_2_ for the prediction of the optical properties of a glass based on its manufacturing parameters, and the realization of more efficient devices. To this end, we here report on the state of the most important intrinsic point defects in pure silica, with a particular emphasis on their main spectroscopic features, their atomic structure, and the effects of their presence on the transmission properties of optical fibers.

## 1. Introduction

The integration of silica glasses in optical and electronic devices are, at present, limited by the effects of high-energy radiation on the transmission and reflection properties of the material. As reported in a large number of studies [1,2,3,4,5,6,7,8,9,10,11], the irradiation of silica with either photons or energetic particles (neutrons, electrons, protons, heavy ions) activates a wide range of damage processes that result in the formation of point defects. These localized irregularities of the network are characterized by one or more energy levels lying in the band gap of the dielectric and are responsible for the shift in the optical absorption edge of the glass to lower energies. The phenomenon is known as photodarkening (or solarization) and is even more pronounced in silica optical fibers (OFs) which, due to their strained glass structure, exhibit a higher sensitivity to radiation as compared to the respective preforms [12,13,14,15,16,17,18,19,20,21,22].

In this review, we analyze the main intrinsic point defects of silica absorbing in the ultraviolet (UV) and visible (Vis) spectral domains and investigate their origin and formation pathways. In the discussion, we adopted a new structure–reactivity approach to understand the difference in the likelihood of defect formation observed in bulk silica and optical fiber silica. In the last part of the review, we explore the consequences that defects have on the transmission properties of optical fibers and how their formation can be prevented. This aspect is particularly important for the development of applications such as fiber laser systems, daylighting systems, and new-generation photovoltaic devices, in which a high transparency in the UV–Vis range is required to guarantee an optimal performance.

## 2. The Geometrical Properties of Silica

SiO_2_ is the chemical formula of a group of minerals constituting about 95% of the Earth’s rocks and soils [23,24]. The most stable polymorph is α-quartz, which has a trigonal crystal structure and exists in both a right- and left-handed form. As shown in Figure 1, when α-quartz is heated to 573 °C at atmospheric pressure, it transforms into β-quartz and assumes a hexagonal crystal structure [25,26]. By further increasing the temperature, β-quartz turns into HP-tridymite (870 °C) and then into β-cristobalite (1470 °C). At 1713 °C, solid β-cristobalite melts into fused silica, which is a homogeneous and isotropic material characterized by a highly randomized structure. Since phase transitions are reversible, a slow cooling of the melt leads back to the crystalline form. However, if the temperature drop is rapid, the system freezes in a metastable state and eventually relaxes to silica. This non-crystalline form of SiO_2_ is characterized by a disordered network lacking long-range order and by the presence of low-density regions.

The building block of all SiO_2_ polymorphs is the [SiO_4_]^0^ tetrahedron (Figure 2), that is, the three-dimensional unit formed by a silicon atom and four oxygen atoms bonded to it. Each O atom bridges two Si atoms and serves as a connection between adjacent tetrahedral units. The Si–O bond has a mixed covalent/ionic character and is described by a wavefunction with both $\sigma [\mathrm{Si}\left(sp^{3}\right)$–O(2p)] and π[Si(3d)–O(2p)] contributions [27,28]. This determines the shortening of the Si–O bond length (1.605 Å), the increase in the Si–O–Si angle, and the variability in the intertetrahedral angles [29]. On the other hand, the O–Si–O bond angle (ϕ) is 109.47°, indicating that the [SiO_4_]^0^ unit has a perfect tetrahedral geometry [30,31].

The relative position of two corner-sharing tetrahedra is determined by the Si–O–Si bond angle (θ) and the Si–O–Si–O and O–Si–O–Si torsion angles. Many experimental works have attempted to estimate θ, but the results greatly vary from study to study. The reason for this uncertainty is that the angle is correlated with the Si–O and Si–Si distances and its value cannot directly be determined from experimental data. In fact, the interference function obtained by X-ray and neutron diffraction techniques only provides the distribution of interatomic distances and the estimation of bond and torsion angles requires the mathematical modeling of the total correlation function [32,33]. Similarly, in solid-state nuclear magnetic resonance (NMR) spectra, the dependence of the ^29^Si chemical shift and the ^17^O quadrupolar coupling on θ requires an accurate correlation function in order to extract the bond angle distribution [34,35,36]. In either case, the structural modeling is generally underconstrained and the reliability of the results depends on the goodness of the starting assumptions [37]. Considering the data reported in the literature and the outcomes of their own simulations, Yuan and Cormack proposed a distribution of Si–O–Si angles with a mean value of 147° and a standard deviation of 10–13° [38]. This result was confirmed by Carpentier et al., who derived a mean value of 147° ± 11° by combining molecular dynamics with first-principles calculations of NMR parameters [33]. More recently, Malfait et al. analyzed a large set of data and located the most probable Si–O–Si angle at 149° with a full width at half maximum (FWHM) of 16° [39]. However, this value only corresponds to the peak of the mean (i.e., averaged over the four corners of each SiO_4_ tetrahedron) Si–O–Si angle distribution derived from different NMR spectra. To calculate the individual Si–O–Si angle distribution, one should first know the statistical distribution of θ at the corners of the tetrahedra. Since a certain degree of correlation among these angles is inevitable, the individual Si–O–Si bond angles are not statistically independent and their distribution cannot be derived from experimental measurements.

Since the publication of Wright’s paper [40], the three-dimensional organization of silica tetrahedra has been discussed in terms of two distinct torsion angles. Using the notation given in Figure 2, the angle $\omega_{1}$ is defined as the dihedral angle between O_1_ (or, equivalently, O_2_ or O_3_), Si_1_, O_4_, Si_2_, while the angle $\omega_{2}$ is defined by the atoms Si_1_, O_4_, Si_2_, O_5_ (or, equivalently, O_6_ or O_7_). However, from a geometrical perspective, the existence of two independent torsion angles is only justified if we define a conventional order to report the structure of silica (as for the primary structure of proteins). Since it is not possible to assign a univocal direction of rotation to silica rings, the two torsion angles are equivalent and carry the same information. For this reason it is sufficient to consider only one of the two torsion angles which we will call ω. From the analysis reported in the literature [38,41], it is found that ω varies as a function of the Si–O–Si bond angle and that, for θ in the 140–160° range, it has three maxima of around 60°, 180°, and 300°. These values correspond to the three staggered conformations of the O_3_Si–O–SiO_3_ moiety, as viewed along the Si–Si axis, and are such that next-nearest-neighbor oxygen atoms are at a maximum distance.

## 3. The Topology of Silica

The analysis of the network topology of silica was first addressed by King, who proposed the shortest-path criterion to identify closed paths of alternating Si–O bonds [42]. According to this model, a ring is defined as the shortest closed path connecting two oxygen atoms bonded to the same Si atom. If the path is formed by *n* bonds (or, equivalently, by 2*n* atoms) the ring is called an *n*-membered ring (*n*-MR) [43,44]. By applying King’s algorithm to α-quartz and β-cristobalite, both polymorphs are composed of regular six-membered rings arranged into different three-dimensional structures [45]. The former network is made up of interconnected hexagonal and trigonal helices of SiO_4_ tetrahedra, whereas the latter consists of stacked layers of tetrahedra alternately pointing up and down [46]. In the case of amorphous silica, the flexibility of the Si–O–Si and Si–O–Si–O angles determines a high degree of structural disorder and this, in turn, leads to a quasi-random orientation of the tetrahedral units and a much broader ring-size distribution [47].

Many theoretical works have been dedicated to the statistical analysis of computer-modeled *a*-SiO_2_ structures. In Figure 3, we report the ring-size distribution of three models obtained by different computational methods. The first dataset was calculated by Rino et al. from a structural model obtained using classical molecular dynamics [31]. The second set comes from first-principle molecular dynamics calculations [48], while the last one is from a model simulated using a combination of classical and first-principles molecular dynamics [49]. The ring statistics of the three models shows that five- and six-membered rings are always the most frequent, followed by seven-membered rings. Three- and four-membered rings, however, have different populations and their concentration depends on the simulation conditions and thermal history of the sample. The prevalence of 6-MRs can be explained by considering the volume–temperature diagram in Figure 4. In fact, when melted silica is cooled at a very slow rate, the atoms in the melt have enough time to rearrange and form crystalline β-cristobalite. Conversely, if the cooling is fast (quenching), the liquid cannot equilibrate with the forming solid and the result is the formation of amorphous silica [50]. Despite their different properties, the two polymorphs ideally derive from the same silica melt and their short- and medium-range topology shows a prevalence of six-membered rings.

The presence of small-sized rings in vitreous silica is strongly correlated with the kinetics of the relaxation process. In fact, when the temperature of the melt is lowered avoiding crystallization, the liquid enters the supercooled phase and its structure continuously rearranges to follow the system temperature. As a consequence, the first-order thermodynamic properties of the liquid (e.g., volume and enthalpy) decrease without any abrupt changes and the viscosity increases accordingly. When the viscosity becomes too high, the atomic motion slows down to the extent that the atoms cannot rearrange themselves into the volume characteristic of that temperature and pressure. At this point, the enthalpy begins to deviate from the equilibrium line and starts to follow a curve of gradually decreasing slope. Eventually, the structure of the system becomes fixed and the supercooled liquid solidifies into silica [51]. The range of temperatures over which the transition occurs is called the glass transformation range and the temperature at which the structure of the supercooled liquid is *frozen-in* in the solid state is the fictive temperature (*T*_f_). Since the departure of the enthalpy from the equilibrium curve is determined by the viscosity of the liquid, i.e., by kinetic factors, a slower cooling rate allows the enthalpy to follow the equilibrium curve to a lower temperature. In this case, the fictive temperature shifts to lower values and the resulting glass has a different atomic arrangement than a more rapidly cooled one [52]. In particular, a slow quenching gives rise to a glass containing mostly medium-sized rings (6- and 7-MRs), whereas an increase in the quenching rate leads to a glass with a greater population of small- and large-sized rings [53].

## 4. Generation of Point Defects

During the formation of a glass, part of the disorder characterizing the supercooled liquid is frozen-in in the solid state and the excess energy is stored as strained Si–O–Si bond angles [32,54]. The presence of distorted angles determines the formation of local high-energy structures like three- and four-membered rings. For regular planar three-membered rings, θ takes the value of 130.5° whereas, for regular planar four-membered rings, its value is 160.5°. In both cases, θ is far from its optimal value (144–155°) and, for this reason, the rings tend to release the excess energy by breaking a strained bond and turn into a bigger ring [55,56].

While in bulk silica the concentration of three- and four-membered rings is relatively low, silica optical fibers exhibit a higher concentration of small rings due to the residual stress generated during the manufacturing process [57]. In fact, the rapid cooling of optical fibers results in a *T*_f_ higher than that measured in bulk samples [58]. This is mainly due to the influence of the drawing speed on the quenching dynamics. A higher drawing speed means a faster cooling rate which, in turn, results in a higher fictive temperature. In addition, the residual stress of the tensile load applied during the drawing process has a strong impact on increasing the concentration of defect precursors in non-irradiated samples [15,16]. This explains the higher sensitivity of OFs to radiation and the higher concentration of room-temperature stable point defects [59,60]. In the remaining part of this section, we will describe the three main intrinsic defects induced by radiation in silica-based materials.

### 4.1. E${}^{\prime }$ Centers

The best-known defect in crystalline SiO_2_ and silica is the *E*${}^{\prime }$ center, that is, an unpaired electron in a Si dangling bond. In their experimental work, Hosono et al. demonstrated that strained bonds contained in three- and four-membered rings absorb F_2_-excimer laser light (7.9 eV, 157 nm) and jump to the lowest electronic spin triplet state (*T*_1_), causing the opening of the ring and the consequent formation of a pair of point defects [53]. The photolytic reaction can be written as
(1)$$\equiv Si–O–Si\equiv ⟶{\left[\equiv Si^{-}{¯¯¯}^{+}O–Si\equiv \right]}^{*}⟶\equiv Si^{•}\;{\;}^{•}O–Si\equiv ,$$
where the three dashes represent three separated Si–O bonds, the asterisk indicates an electronically excited state, and the dot represents an unpaired electron. The first step of the reaction involves the transition of an electron from the valence to the conduction band and the formation of a self-trapped exciton (STE) consisting of an excited electron (*e*^–^) located on a silicon atom and a hole (*h*^+^) trapped at one or more neighbouring oxygen atoms [61,62]. This structure is kept metastable by the spontaneous creation of a localised distortion in the SiO_2_ network, which lowers the total energy of the system and traps the *e*^–^–*h*^+^ pair at the distortion site [63]. Using self-consistent quantum chemical calculations, Shluger proposed a model in which the exciton self-trapping is accompanied by the weakening of a Si–O bond and the displacement (0.3 Å) of the oxygen atom towards an interstitial position [64]. Other calculations showed that, after electronic excitation, the system relaxes to the nearest energy minimum on the excited-state energy surface by breaking a Si–O bond and moving the Si atom into a planar *sp*^2^ configuration [62,63]. In either case, the relaxation of the STE to the ground state (and the possible rehybridization of the silicon orbitals to an *sp*^3^ configuration) leads to the formation of a silicon and an oxygen dangling bond known, respectively, as *E*${}_{\alpha }^{\prime }$ center and non-bridging oxygen hole center (NBOHC) [65].

A second pathway for *E*${}^{\prime }$ center formation was proved by Tsai and Griscom while studying the effect of highly-focused ArF excimer laser light (6.4 eV, 193 nm) on silica specimens [66]. As in the previous case, the first step of the proposed mechanism consists in the promotion of an electron to the conduction band and the formation of a self-trapped exciton. Here, however, its de-excitation proceeds through the displacement of the oxygen atom and the formation of an interstitial O^0^ atom and a ≡Si–Si≡ oxygen vacancy (Frenkel defect pair) [67].
(2)$$\equiv Si–O–Si\equiv ⟶{\left[\equiv Si^{-}{¯¯¯}^{+}O–Si\equiv \right]}^{*}⟶\equiv Si–Si\equiv +O^{0}$$

The process involves a non-radiative decay of the electronic excited state, and its efficiency is higher in densified silica where the mean Si–O–Si angle is lower and the mean Si–O bond length is higher [68,69]. By further exciting the oxygen vacancy, the system evolves towards the formation of a Si dangling bond and a nearly planar $\equiv Si^{+}$ unit [70,71]. To distinguish this variant of *E*${}^{\prime }$ center from that obtained via the mechanism (Equation 1), we label it an *E*${}_{\gamma }^{\prime }$ center.
(3)$$\begin{array}{c} \equiv Si–Si\equiv ⟶\equiv Si^{•}\;{\;}^{+}Si\equiv +e^{-} \end{array}$$

Although the major channel for the formation of *E*${}^{\prime }$ centers has not yet been identified, it appears that the bond-dissociation mechanism (Equation 1) prevails at higher irradiation energies whereas the Frenkel-type mechanism (Equation 2) and (Equation 3) predominates at lower energies. This view is supported by experimental results showing that the concentration of *E*${}_{\alpha }^{\prime }$ and NBOHC defects linearly increase with the pulse energy of F_2_ lasers (7.9 eV) [53], while that of *E*${}_{\gamma }^{\prime }$ centers and O^0^ quadratically increases with the pulse energy of KrF (5.0 eV) and ArF lasers (6.4 eV) [72,73]. As a consequence, it has been suggested that the rupture of the Si–O bond described by Equation (Equation 1) is assisted by the absorption of a single photon [74], whereas the cleavage of the O atom and the successive formation of the *E*${}_{\gamma }^{\prime }$ center is induced by a two-photon absorption process [66]. In the latter case, two alternative pathways have also been proposed. If the absorption is not simultaneous, the first photon is responsible for the formation of the STE while the second serves to ionise the oxygen vacancy and produce the Si dangling bond [75]. These two stpdf correspond to reactions (Equation 2) and (Equation 3) and each of them is activated by the absorption of one photon. Contrarily, if the two photons are absorbed at the same time, the reaction goes through a biexciton process in which one of the two STEs decays as reported in Equation (Equation 2), while the other supplies a hole to foster the reaction [76]: (4)$$\begin{array}{c} \equiv Si–Si\equiv +h^{+}⟶\equiv Si^{•}\;{\;}^{+}Si\equiv . \end{array}$$

Two additional variants of *E*${}^{\prime }$ defects have seldom been observed in irradiated silica. *E*${}_{s}^{\prime }$ is a hemi-center typically observed on silica surfaces or interfaces, comprising only a threefold coordinated Si atom with the unpaired electron. The *E*${}_{\beta }^{\prime }$ center, on the other hand, features a $\equiv Si^{•}$ moiety coupled with a hydrogen-saturated oxygen vacancy [77,78]. This defect is generally found in hydrogen-rich silica, where the concentration of silanol groups (SiOH) is higher. In fact, the irradiation of these glasses with F_2_ lasers or ionizing beams leads to the rupture of the O–H bond and the formation of a NBOHC and a neutral hydrogen atom (H^0^) as follows: (5)$$\begin{array}{c} \equiv SiOH⟶\equiv SiO^{•}\;+H^{0}. \end{array}$$

The free hydrogen may then diffuse in the silica matrix and react with a pre-existing ≡Si–Si≡ oxygen vacancy to generate an *E*${}_{\beta }^{\prime }$ center [79]: (6)$$\begin{array}{c} \equiv Si–Si\equiv +H^{0}⟶\equiv Si^{•}\;\;H–Si\equiv . \end{array}$$

Since all the *E*${}^{\prime }$ variants have very similar electronic structures, their optical absorption (OA) spectrum is characterized by the same band, peaking at about 5.8 eV (214 nm) with a full width at half maximum (FWHM) of 0.8 eV and an oscillator strength *f* = 0.14 ± 0.1 [80,81]. An experimental OA spectrum obtained after γ-ray irradiation of a wet silica sample is reported in Figure 5. The electronic transition corresponding to this band is still debated. One hypothesis is that the OA originates from the charge transfer from valence band states (i.e., a 2*p* orbital of one of the three O atoms bonded to the *E*${}^{\prime }$ center) to the empty state of the Si dangling bond [82]. This is supported by density functional theory calculations showing that the lower part of the absorption spectrum of *a*-SiO_2_ corresponds to the superposition of the O(2*p*)→Si(*sp*^3^) transition with that from the occupied Si(*sp*^3^) state to the diffuse states in the lower part of the conduction band [83].

### 4.2. Non-Bridging Oxygen Hole Centers

The non-bridging oxygen hole center (NBOHC) is the simplest oxygen-related intrinsic defect in silica. It corresponds to a Si atom bonded to an O atom having a dangling bond, i.e., an unpaired electron in a 2*p*-like non-bonding orbital [65]. As mentioned above, there are two major pathways through which NBOHCs can be created. The first is called the intrinsic mechanisms and consists of the photolysis of a strained Si–O–Si bond, as given in reaction (Equation 1). The second is the extrinsic mechanism, which corresponds to the homolytic dehydrogenation of a silanol group bond, shown in reaction (Equation 5). The predominance of one mechanism over the other is determined by the presence of pre-existing defects, their concentration, and the energy of the excitation beam [85].

The electronic structure of NBOHCs was fully described by Suzuki et al. using *ab initio* cluster calculations [86]. The proposed energy level diagram is illustrated in Figure 6. Apart from the bonding and antibonding σ orbitals, the diagram shows a series of O non-bonding orbitals whose degeneracy is lifted by the interaction of the defect with the surrounding atoms of the amorphous network. In particular, the interaction splits the non-bonding orbitals of the bridging oxygens into two sets of multiply-degenerate *n_p_x__*(O_B_) and *n_p_z__*(O_B_) orbitals, and the lone-pair orbitals of the non-bridging oxygen into two *n_p_x__*(O_NB_) and *n_p_y__*(O_NB_) levels. The highest occupied molecular orbital (HOMO) is the singly-occupied *n_p_y__*(O_NB_) orbital whereas the lowest unoccupied molecular orbital (LUMO) coincides with the antibonding $\sigma_{p_{z}}^{*}$(Si-O_NB_) orbital. The promotion of an electron from the ground-state levels to the HOMO gives rise to three OA bands corresponding to three distinct electronic transitions [87,88]:(i)An asymmetric Pekarian-shaped band peaked at 1.97 eV (FWHM = 0.17 eV, *f* ≈ $1.5\times 10^{-4}$ eV) attributed to the $\sigma_{p_{z}}$(Si-O_NB_)→*n_p_y__*(O_NB_) transition from the bonding σ orbital to the half-filled orbital of the non-bridging O atom (the HOMO of the cluster).(ii)A band centered at 4.8 eV (FWHM = 1.07 eV, *f* ≈ 0.05) originating from the *n_p_y__*(O_B_)→*n_p_y__*(O_NB_) transition between the O_B_ lone-pair orbital perpendicular to the Si–O–Si plane and the HOMO.(iii)A band at 6.8 eV (FWHM ≈ 1.8 eV, *f* = 0.05) related to the *n_p_x__*(O_B_)→*n_p_y__*(O_NB_) transition from the O_B_ lone-pair orbital lying in the Si–O–Si plane to the HOMO.

The decay of the excited state created by the above transitions gives rise to a photoluminescence (PL) band at 1.91 eV with an FWHM = 0.17 eV and a lifetime of ∼14 μs [80,89]. As was generally expected, the excitation from the σ orbital to the HOMO should imply a weakening of the $Si–O^{•}$ bond and a lengthening of the mean bond distance. Instead, experimental results indicate that the frequencies of the $Si–O^{•}$ symmetric stretching mode in the ground (890 cm${}^{-1}$) and the excited state (860 cm${}^{-1}$) are almost the same and the Stokes shift between the excitation and emission bands is as small as 0.06 eV [90,91,92]. This anomalous behaviour is caused by the interaction of the doubly occupied *n_p_y__*(O_NB_) orbital in the excited state with a symmetry-adapted combination of the three empty $\sigma_{p_{y}}^{*}$(Si-O_B_) orbitals [86]. This so-called "negative hyperconjugation" is responsible for the partial delocalization of the electron density from the oxygen to the silicon atom and for the resulting stabilization of the $\sigma_{p_{z}}$(Si-O_NB_) MO in electronically excited NBOHCs. In this way, the small electron–phonon coupling typical of NBOHC and the almost-equal $Si–O^{•}$ bond length in the ground and excited state can be explained.

### 4.3. Oxygen-Deficient Centers

Oxygen-deficient centers (ODCs) are the basic type of neutral oxygen monovacancies in non-stoichiometric silica and generally correspond to a Si–Si dimer configuration [93]. They are naturally present in unirradiated silica but their concentration considerably rises when a glass is irradiated with UV, X-ray, or γ-ray beams [80]. The major formation pathway of ODCs is given in Equation (Equation 2), where an energetic photon causes the release of an interstitial oxygen atom from the silica network to form a Si–Si bond.

The first spectroscopic studies on ODCs were performed in the mid-1950s by Garino-Canina [94], Mitchell and Paige [95], and Cohen [96], among others. Their results led to the identification of two optical absorption bands called "E-band" (7.6 eV) and "B_2_-band" (5.0 eV), which were tentatively assigned to interstitial oxygen atoms and divalent silicon atoms, respectively. In addition, three photoluminescence bands (called α, β, and γ) were observed at approximately 4.3 eV, 3.1 eV, and 2.7 eV, and associated with substitutional Ge atoms at oxygen-vacancy sites. In 1983, O’Reilly and Robertson [97] calculated the electronic structure of the main defects in SiO_2_ and suggested two different variants for the oxygen-deficient center. The so-called ODC(I) was proposed to be a relaxed ≡Si–Si≡ oxygen vacancy, while ODC(II) was identified with an unrelaxed ≡Si–Si≡ bond of length 3.06 Å. They also calculated the energy levels for both structures and demonstrated that the 7.6 eV OA band could be associated with the σ→$\sigma^{*}$ transition of the relaxed Si–Si bond. This picture was partially confirmed by Hosono et al. [98] and Imagawa et al. [99] who found that, by heating unirradiated SiO_2_ glasses in either hydrogen or oxygen gas flow, the intensity of the E-band decreased in accordance with the assumption that H_2_ and O_2_ neutralize pre-existing ODCs, as shown by
(7)$$\begin{array}{c} \equiv Si–Si\equiv +H_{2}⟶\equiv Si–H\;\;H–Si\equiv \end{array}$$
and
(8)$$\begin{array}{c} \equiv Si–Si\equiv +\frac{1}{2}O_{2}⟶\equiv Si–O–Si\equiv . \end{array}$$

These reactions provided the definitive proof of ODC(I)’s structure and optical activity, but did not clarify the nature of the other defect variant. As experimental evidence built up, two alternative structural models were put forward to explain the spectroscopic behavior of ODCs(II). One is the neutral oxygen vacancy ($V_{O}^{0}$) model originally proposed by O’Reilly and Robertson [97] and subsequently adopted by Imai et al. [75,100] to interpret their findings. While irradiating dehydrated oxygen-deficient silica with ArF and KrF excimer laser, they observed a non-linear decrease in both the B_2_-band and the PL α-band, accompanied by a dose-dependent generation of *E*${}^{\prime }$ centers. The intensity of the E-band, however, did not show any change during the experiments and remained at the original intensity level. The analysis of the concentration of *E*${}^{\prime }$ with irradiation time revealed that the growth curve can be decomposed into an initial, saturating part due to ODC(II) and a larger, linear-growth component attributable to ODC(I). The formation mechanism of *E*${}^{\prime }$ centers was thus proposed to proceed via the direct photoionization of ODCs or hole-trapping, as given in reactions (Equation 3) and (Equation 4). The different response of the 7.6 eV and 5.0 eV bands to laser irradiation can be explained by assuming that the formation efficiency of *E*${}^{\prime }$ from ODC(II) is much higher than that from ODC(I), probably because of the similarity between the unrelaxed oxygen vacancy and the $\equiv Si^{+}$ atom accompanying the *E*${}_{\gamma }^{\prime }$ center.

The second model, called the twofold-coordinated silicon (${\mathrm{Si}}_{2}^{0}$) model, was proposed by Skuja to interpret the origin of the OA band at 5.0 eV and the PL bands at 4.3 eV and 2.7 eV [101]. In the ${\mathrm{Si}}_{2}^{0}$ notation, the “2” stands for the coordination number of the Si atom and the “0” for its net electric charge. The structure proposed for the ODC(II) was that of a divalent silicon bonded to two bridging O atoms and with a lone pair in an *sp*^2^ orbital. The transition that gave rise to the B_2_-band was identified with the S_0_→S_1_ excitation of the twofold Si atom, while those associated with the α- and γ-bands were the transitions S_1_→S_0_ and T_1_→S_0_ at the same defect site. This model was supported by the studies of Tsai and Griscom on the structure of a hydrogenated variant of the *E*${}^{\prime }$ center, called an H(I) center [102]. They demonstrated that the electron paramagnetic resonance (EPR) features of this defect were compatible with an *sp*^3^ silicon atom bonded to two oxygens and one H atom, and resumed the pathway described by Radtsig [103] as a formation mechanism: (9)$$\begin{array}{c} \equiv Si–O^{•}+H_{2}⟶\equiv Si–OH+H^{0} \end{array}$$
(10)$$\begin{array}{c} =Si:+H^{0}⟶=Si^{•}–H. \end{array}$$

The first step corresponds to the annealing of an NBOHC to give a silanol group and a hydrogen atom. The second step is the reaction of latter with an ODC(II) to convert it into an H(I) center. Both reactions have been confirmed in a number of experimental and theoretical studies and represent the cornerstone of the ${\mathrm{Si}}_{2}^{0}$ model [104,105,106,107].

In 1989, the ODC(II) issue got even more complicated when Tohmon et al. reported the existence of two accidentally overlapping bands making up the B_2_-band [108]. The first contribution (called B_2α_) was centered at 5.02 eV (FWHM=0.35 eV) and was related to two emission bands at 4.42 and 2.7 eV. The other band (B_2β_) was peaked at 5.15 eV (FWHM=0.48 eV) and was linked to emission at 4.24 eV and 3.16 eV. By analyzing the decay lifetime of the photoluminescence, the authors proposed that the B_2α_-band corresponds to the singlet-to-triplet (S_0_→T_1_) transition of the relaxed Si–Si bond while the corresponding luminescence is due to the inverse transition T_1_→S_0_. The origin of the B_2β_-band was not discussed by the authors but it was subsequently attributed by Kohketsu et al. [109] to the =Si**:** center, together with the corresponding 4.24 and 3.16 eV PL bands. These assignments were largely criticized mainly because the B_2α_-band lifetime (τ≈100 μs) did not match the value of 10 ns reported in many other studies [110,111,112]. Moreover, Anedda et al. observed that the 4.4 eV emission band was due to an intrinsic defect and that it shifted to 4.2 eV when the defect was perturbed by an unidentified impurity [113]. The two PL bands were thus called α_*I*_ and α_*E*_, where *I* stands for intrinsic and *E* for extrinsic. On the basis of these observations, Skuja re-elaborated the ${\mathrm{Si}}_{2}^{0}$ model including the possibility for Ge and Sn to form ODC-like defects which could contribute to the optical properties of low-purity and Ge-doped SiO_2_ glasses. According to this new $T_{2}^{0}$ model (with *T* standing for Si, Ge, or Sn), the divalent =Si**:** center is responsible for the OA band peaked at 5.02 eV (S_0_→S_1_ transition) [112,114], as well as for the PL α_*I*_ (S_1_→S_0_) and β (T_1_→S_0_) bands [101,115]. Similarly, the isoelectronic =Ge**:** defect gives rise to the 5.15 eV OA band and the two PL bands α_*E*_ and γ (transitions as before). This model has gained a wide acceptance in recent years and is backed up by both theoretical [116,117,118,119] and experimental [120,121,122] investigations.

## 5. Photodarkening in Optical Fibers

When silica optical fibers are exposed to ionizing radiation, some of the strained chemical bonds present in the core are broken by the incoming light to give rise to point defects. The microscopical damages are primarily manifested as a degradation of the optical signal-to-noise ratio and a decrease of the optical power along the waveguide. As discussed above, the glass found in OFs is more sensitive than bulk silica due to its higher content of small-sized rings. This is determined by the particular conditions encountered during the manufacturing of the fibers (i.e., higher quenching rate, drawing speed, and applied strain) and is reflected in the greater propensity of OFs to undergo photodarkening. Furthermore, the glass stoichiometry, the content of hydroxyl groups (OH), and the concentration of impurities also impact the radiation sensitivity of a optical fibers.

Radiation-induced losses in silica OFs have been shown to be caused primarily due to Si-related defects or to absorbing species related to chlorine impurities. The most important contribution is given by *E*${}^{\prime }$ centers originating either from oxygen-deficient centers, extrinsic Si–H bonds, or strained Si–O bonds [15,16,17,123]. When *E*${}^{\prime }$ centers come from pre-existing defects (i.e., ODCs), the growth curve observed under irradiation shows a saturating profile due to the exhaustion of precursors. Conversely, when they are generated from the photo-assisted dissociation of strained bonds, the growth kinetics of *E*${}^{\prime }$ centers is linearly correlated with that of NBOHCs and both have the same dependence on the dose of radiation received by the material [123]. In many case, the decomposition with Gaussian absorption bands of the radiation-induced attenuation spectra measured for irradiated OFs reveals the presence of ODC(I)s and ODC(II)s at lower concentrations than *E*${}^{\prime }$ and NBOHCs [124]. At greater energies, the addition of two absorption bands centered at 3.26 eV (FWHM = 1.2 eV) and 3.78 eV (FWHM = 0.57 eV) is often necessary in order to improve the quality of the fit. These two bands have been associated with chlorine species (namely, Cl${}^{0}$ and Cl${}_{2}$) deriving from the detachment of Cl atoms from the silicon tetrachloride used to make optical fibers [124,125]. Since all these defects absorb light in the 180–700 nm spectral window, the radiation-induced attenuation is usually higher in the UV–Vis domain and lower in the near-infrared range (700 to 2000 nm). This determined that optical fibers have historically been used in telecommunication and sensing systems to transmit signals in the infrared domain [126,127,128,129]. However, the recent technological advances in fields like UV photolithography [130,131], laser surgery [132,133,134], and fiber-based photovoltaics [135,136,137,138,139,140,141] have given a new impulse towards the development of waveguides able to withstand the damaging effects of high-energy photons and maintain a high transmittance of UV and optical signals.

Radiation-hardened optical fibers are specialty waveguides whose composition has been optimized to strongly reduce their radiation sensitivity. Hydrogen loading of pure silica-core OFs was shown to considerably increase the resistance to ionizing radiation and, at the same time, decrease the intrinsic attenuation level at 700 nm [142,143,144,145]. In fact, the presence of molecular hydrogen in the silica matrix can help the recovery of optical fibers by annealing photo-induced *E*${}^{\prime }$ centers and NBOHCs as follows [145]: (11)$$\begin{array}{c} \equiv Si^{•}+H_{2}⟶\equiv Si–H+H^{0} \end{array}$$
(12)$$\begin{array}{c} \equiv Si–O^{•}+H_{2}⟶\equiv Si–OH+H^{0}. \end{array}$$

However, it has been shown that most of the hydrogen quickly diffuses out of the core material and only a “stable” amount of H_2_ remains in the silica matrix [146]. This means that the anti-radiation efficiency of these fibers depends on the time interval between treatment and irradiation, and that their use should be limited to applications requiring only short-term stability [147,148,149]. To increase the useful life of H_2_-loaded fibers, hermetic coatings made of carbon or metal have been developed to prevent the diffusion of hydrogen out of the fiber [150,151]. This technique was shown to improve the resistance to prolonged irradiation but the strict conditions required to deposit the coating material on the surface of silica still prevent its use in common devices [19]. More recently, Hartung et al. designed a so-called anti-resonant hollow-core fiber (AR-HCF), which is able to guide light in three transmission bands in the UV region with minimum attenuation (1 to 10 dB·m${}^{-1}$) [152]. Unlike refractive-index guiding fibers, this particular type of waveguide features a microstructured cladding enveloping an air/vacuum core, which guides light via the anti-resonant reflection optical waveguiding mechanism [153,154]. Due to their low modal overlap factor, AR-HCFs usually exhibit high laser damage threshold and high radiation resistance, as well as low modal dispersion, low material absorption, and low optical nonlinearity. All these characteristics make anti-resonant hollow-core fibers good candidates for challenging applications such as high-power UV laser, nonlinear and ultrafast optics, plasma physics, and surgery [155,156,157]. Nonetheless, their relatively high cost and low availability in the market still favor the use of more economic silica-based OFs for common applications.

## 6. Summary and Future Directions

In this review, we investigated the structural and optical properties of the most common UV–Vis absorbing point defects found in pure silica. We have focused our attention on the relationship between pristine glass structure and the photo-induced generation of *E*${}^{\prime }$ centers, NBOHCs, and ODCs. By analyzing the short-range properties of *a*-SiO_2_, we observed that the spatial arrangement of the silicon and oxygen atoms can be described in terms of the Si–O bond length (1.605 Å), the O–Si–O (ϕ) and Si–O–Si (θ) bond angles, and a single Si–O–Si–O dihedral angle (ω). The amplitude of ϕ is fixed to 109.47° by the rigid tetrahedral geometry of the [SiO_4_]^0^ unit, whereas the value of θ follows an asymmetric distribution that peaked at approximately 147° with an FWHM of ∼16°. The torsion angle ω was found to be strongly correlated with the Si–O–Si angle and, for 140°≤θ≤160°, it takes the value 60°, 180°, and 300°. Traditionally, the arrangement of adjacent SiO_4_ tetrahedra has been described in terms of two distinct torsional angles defined by the sequences of consecutive atoms O–Si–O–Si and Si–O–Si–O. However, since these sequences are not ordered in silica, the two angles are equivalent and the three-dimensional properties of the network can be described by using just one of them.

Due to the flexibility of the angles θ and ω, the tetrahedral SiO_4_ units arrange themselves to form closed structures called rings. The most abundant rings are composed of six tetrahedra, followed by those constituting five and seven units. Rings of three and four units are characterized by high steric and angular hindrance and, for this reason, they have a high free-energy content. Although their concentration in natural silica is relatively low, glasses obtained by melt quenching (such as those of optical fibers) show an increased concentration of small-sized rings due to the freezing-in of unrelaxed local structures typical of the supercooled liquid state. This feature has important consequences for the optical properties of the glass and its resistance to ionizing radiation. In fact, as we have pointed out in this review, the atoms contained in three- and four-membered rings have a higher propensity to interact with the incoming radiation and release excess energy by breaking a Si–O bond and forming a larger ring.

The most straightforward example of the photo-assisted cleavage of Si–O bonds is the formation of *E*${}^{\prime }$ centers. The first step of the reaction is the absorption of a photon by a strained bond and the formation of a metastable self-trapped exciton (STE). Depending on the irradiation energy and the structural properties of the glass, the STE can decay to form either an *E*${}_{\alpha }^{\prime }$–NBOHC pair or an ODC plus a free O${}^{0}$ atom. By further exciting the ODC, the Si–Si bond is ionized and the system evolves towards the formation of an *E*${}_{\gamma }^{\prime }$ center. Two additional variants of *E*${}^{\prime }$ have also been observed on the surface of silica samples (*E*${}_{s}^{\prime }$) and in irradiated wet silica (*E*${}_{\beta }^{\prime }$). Despite their different chemical make-up, all *E*${}^{\prime }$ centers share the same electronic structure and are characterized by a similar OA band, peaking at 5.8 eV. The formation reaction of *E*${}_{\alpha }^{\prime }$ shown in Equation (Equation 1) and that of *E*${}_{\beta }^{\prime }$ shown in Equation (Equation 5) are also considered the two major pathways for the formation of NBOHCs in silica. Due to the complex electronic structure of this defect, its absorption spectrum features three distinct OA bands centered at 1.97 eV, 4.8 eV, and 6.8 eV. The transitions assigned to these bands originate from different initial levels but have the same oxygen non-bonding orbital as a final level. When the excited state created by these transitions decays, the emitted photons give rise to a PL band peaking at 1.91 eV. The first step of the *E*${}_{\gamma }^{\prime }$ formation mechanism Equation (Equation 2) is also the major channel for the photo-induced generation of the color center called ODC(I). The structure of this defect has been largely debated but there is an almost universal consensus at present, recognizing it as a ≡Si–Si≡ dimer configuration with an OA band at 7.6 eV. A second variant of oxygen-deficiency center known as ODC(II) has been observed in a large number of experimental works, but its structural features remain controversial. A first model elaborated in the 1980s treated this defect as an unrelaxed oxygen vacancy consisting of an elongated Si–Si bond. However, this model failed to justify a number of successive experimental evidences and, for this reason, it was partially discarded. The newest and most widely adopted model is that proposed by Skuja, which relates the ODC(II) with a twofold coordinated silicon. The optical activity of this defect was correctly correlated with an OA band centered at 5.02 eV and with a strong emission at 4.42 eV and a very weak emission at 2.7 eV. Moreover, the model accounted for the existence of Ge- and Sn-based oxygen-deficient centers, contributing to the optical spectra of impure silica glasses. In this case, the 5.15 eV OA band as well as the 2.7 and 4.24 eV PL bands are considered to originate from electronic transitions at =Ge**:** centers isoelectronic to ODC(II).

As all the analyzed point defects have absorption bands in the 180–700 nm portion of the spectrum, it is clear that transmitting UV–Vis light in optical fibers in a harsh environment is a very challenging task. To mitigate the photodarkening effects of energetic photons, a large plethora of specialty fibers have been developed in the last two decades. The first and most widespread solution is represented by H_2_-loaded optical fibers, that is, waveguides that have been infused with molecular hydrogen. The presence of the gas in the silica matrix act as a buffer towards the formation of certain point defects by annealing them and keeping the attenuation levels low. To prevent the out-flowing of hydrogen from the core, H_2_-loaded fibers have also been hermetically coated with carbon or metals with a specific high-temperature, high-pressure process. In 2014, a new type of optical fibers, called anti-resonant hollow-core fibers, have been developed for the transmission of high-energy pulses in the UV–Vis domain. These waveguides are not silica-based and allow for the transmission of signals in the core via the anti-resonant effect with a very low attenuation. Nonetheless, their low availability in the market and the high production costs still prevent their diffusion in common devices. For this reason, it is important to further optimize silica optical fibers by controlling the fabrication process parameters to manipulate the nature and concentration of the point defects that are responsible for their degradation.

## Figures and Tables

**Figure 1 materials-14-07682-f001:**
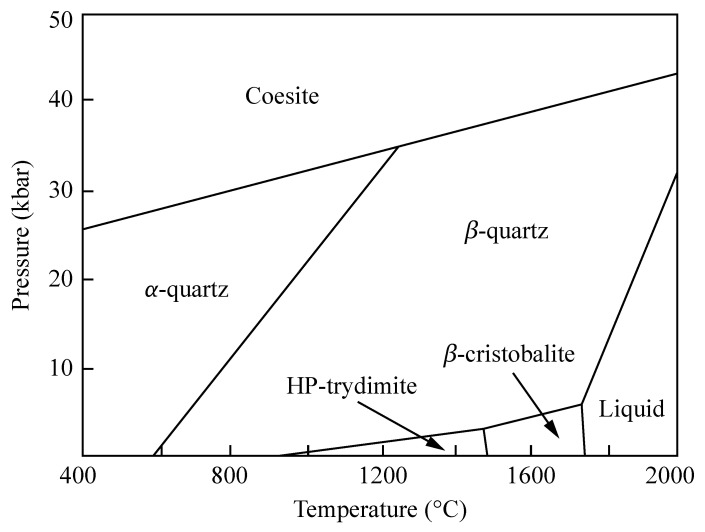
Phase diagram of SiO_2_ showing its main crystalline forms as well as the liquid (melted) phase.

**Figure 2 materials-14-07682-f002:**
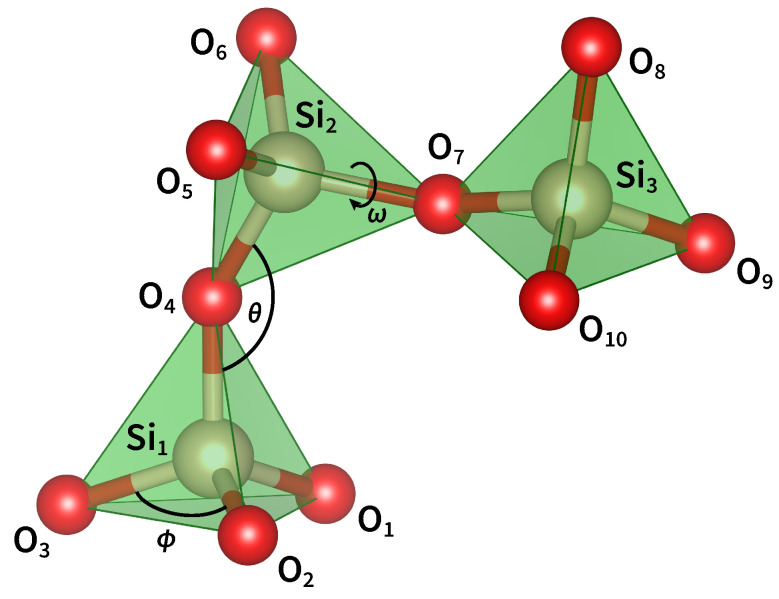
Parameters defining the topology of silica. The angle ϕ corresponds to the O–Si–O bond angle, θ corresponds to the Si–O–Si bond angle, while ω is the dihedral angle between the planes formed by atoms O–Si–O and Si–O–Si.

**Figure 3 materials-14-07682-f003:**
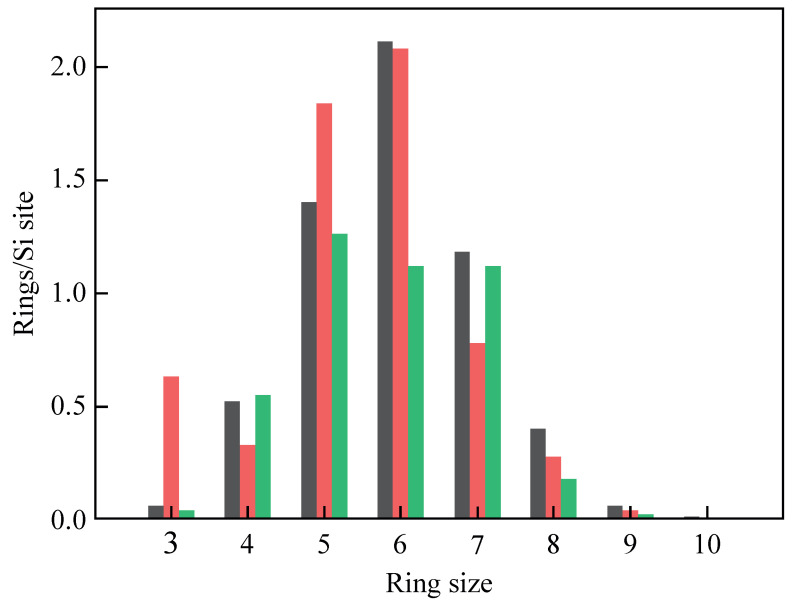
Shortest-path ring statistics of the silica models reported by Rino et al. (grey) [31], Pasquarello and Car (red) [48], and Giacomazzi et al. (green) [49].

**Figure 4 materials-14-07682-f004:**
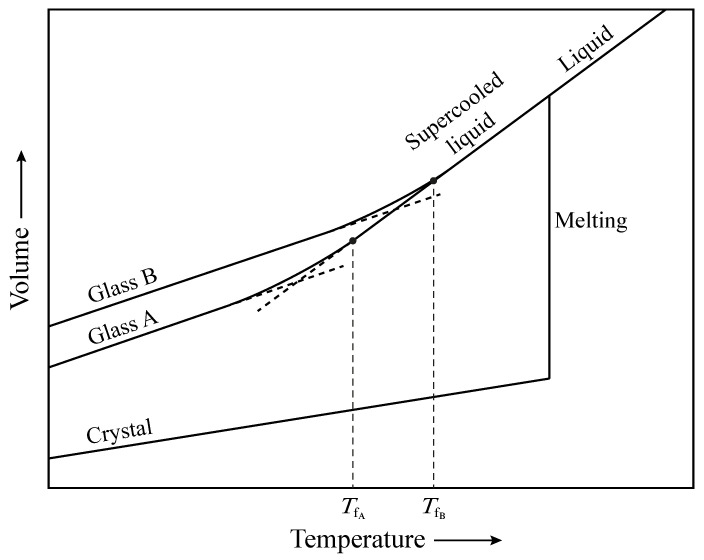
Volume–temperature diagram representing the solidification of melted silica into crystalline or glassy SiO_2_. The supercooled liquid can be cooled at different quenching rates to produce glasses with varying fictive temperatures.

**Figure 5 materials-14-07682-f005:**
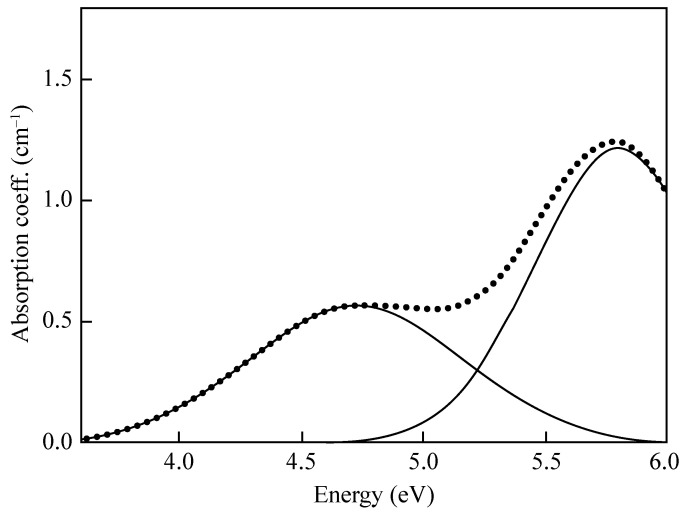
Optical absorption spectrum of a synthetic silica sample showing a band assigned to the *E*${}^{\prime }$ center (5.8 eV) and another band assigned to NBOHC (4.8 eV). Adapted from Cannas et al. [84].

**Figure 6 materials-14-07682-f006:**
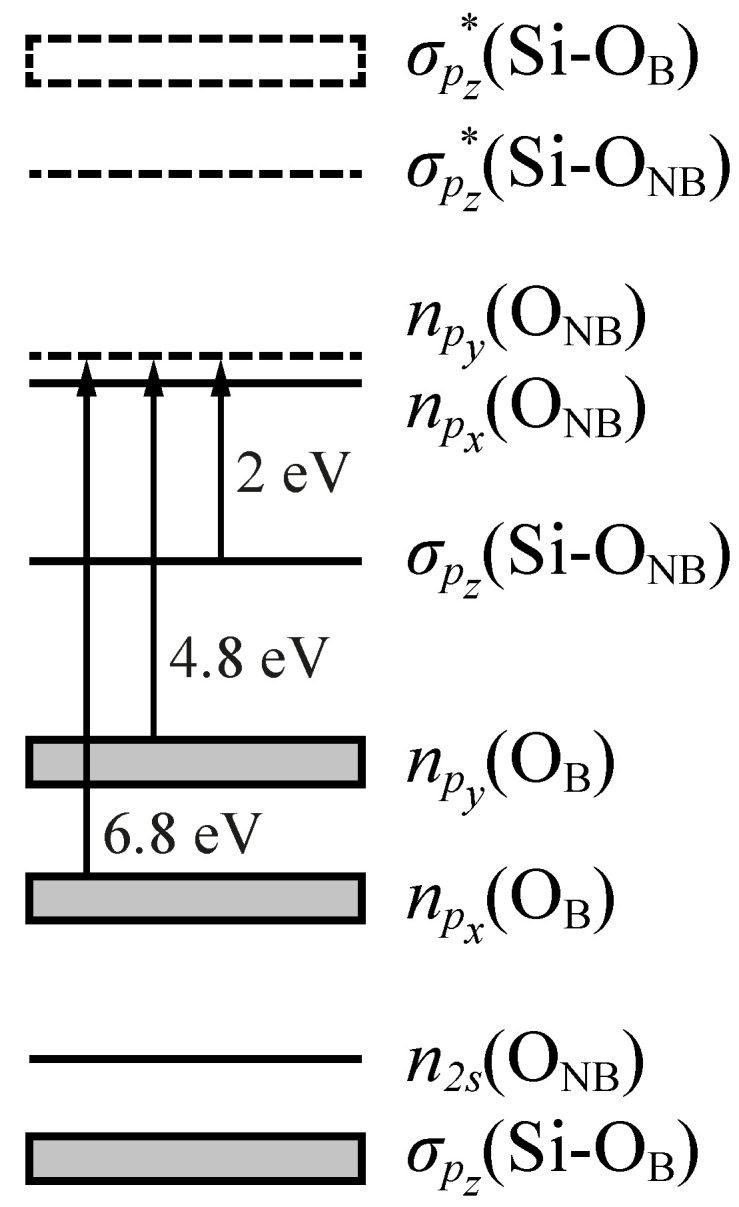
Energy level diagrams of a non-bridging oxygen hole center. Vertical arrows correspond to optical transitions between bondind and non-bonding orbitals, while grey boxes represent multiply degenerated levels. Adapted from Suzuki et al. [86].

## Data Availability

Not applicable.

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
