# Peer review of "Intrinsic Point Defects in Silica for Fiber Optics Applications"

_materials, 2021, doi:10.3390/ma14247682_

Round 1
Reviewer 1 Report
I have to say that this is a good review that focused on the structural and optical properties of UV–Vis absorbing point defects of pure silica. Especially they pay attention to the relationship between pristine glass structure and photo-induced generation of defect center. However, I did not see the latest advance of the subject in the field. Most of the cited literature are very old. Moreover, the author should point out the difficulty, challenge and future development in this field. This is important for a good review.
Author Response
We thank the referee for the positive comment on our manuscript. To take in due account the referee’s suggestions, we have expanded the introduction and the summary, including some hints on the perspectives and an updated literature context. We have also added a whole section called “Photodarkening in optical fibers” where we present the latest development of radiation-hardened OFs.
Reviewer 2 Report
This paper is a review on points defects in silica glass used for optic al fibers.
There are two parts in its organization: a first parts related to the glass structure and a second one related to point defects.
The first parts is valuable making the point on the very recent progress in quantum calculation, the second part on point defects introduces moderate information. Specifically, not all the previous reviews are cited e.g. Griscom, D. L. (1986). Point defects in amorphous SiO2: what have we learned from 30 years of experimentation. Mat. Res. Soc. Symp., Materials Research Society.
Author Response
We thank the referee for his valuable comments.
In order to comply with his/her suggestions, we have reviewed the text and expanded the bibliography with numerous citation of previous reviews and more recent articles regarding the applications of optical fibres in the UV-Vis domain.
Reviewer 3 Report
The paper is not in mdpi format, Materials. The authors translate into the format required in the guide. Didn't the authors consult the guide?
The paper does not have a normal structure for an ISI paper, divided into Introduction, Experimental, Results, Discussions, Conclusions. References are not in the required format.
You don't see your own contribution. Information is extracted from the works. It is not possible to identify the purpose and what it brings in addition to the specialized literature.
I recommend to rewrite the paper in the format required by the journal, to divide it into the chapters required by the magazine, to highlight the personal contribution, originality and degree of novelty. To be more orderly.
Author Response
We are grateful to the referee for the comments.
The proposed paper is not an experimental article but a review focusing on those point defect which can pose a problem for utilisation of optical fibre in the UV-Vis-NIR spectral domain.
The text is written using the latex template provided by MDPI and is formatted accordingly to the recommendations of the journal.
We invite the referee to read more carefully the review.
Reviewer 4 Report
The article is very interesting and discusses the defects occurring in optical fibers, but not mentioned in the publication about any applications of these optical fibers, therefore, in the opinion of the reviewer, the word "aplication" should be removed from the title.
In the article, the authors describe the defects in optical fibers, their types and nature, but they do not refer to the possibilities of using these defects (positive or negative) in practical applications. On the other hand, the title would suggest that their practical applications will be discussed.
Author Response
We thank the referee for the positive and valuable comments, which helped us to improve our manuscript. We have modified the text by adding a selection of very recent articles on the applications of optical fibres in the UV-Vis domain.
We have also added a paragraph discussing the advantages and disadvantages of using photodarkening-prone silica fibres in optical devices.
Reviewer 5 Report
The authors present an excellent introduction to radiatively generated defects in amorphous silica, describing the geometry of the SiO2 structure and listing the defects engendered. The paper is well written, simple, easy to read and highly informative. I believe this paper can be published as is and will make some optional recommendations and corrections below from the perspective of a reader no too familiar with the body of work.
Recommendations:
1) I would recommend writing a paragraph describing the context in which the defects are studied. I would be interested in understanding if the current concerns are due to defects generated during processing or generated during the usage/operation of the material? Under what circumstances do these defects impede the utility of the material?
2) A conclusion paragraph would be very useful to tie everything together, particularly with regards to potential directions in this interesting field of study.
Corrections:
1) In the first sentence of the first paragraph of Section 3, I am confused by the phrase "characterising in", I believe the "in" is unnecessary.
2) In the second last sentence of the second paragraph of Section 3, "sensibility" should be changed to "sensitivity".
3) The phrase "is that given in Eq. (2)" in the first paragraph of the section titled "Oxygen-deficient centres" should be changed to "that is given in Eq. (2)".
Author Response
We are grateful to the referee for his positive evaluation. In the following we will answer to each of his recommendations and corrections.
-
As explained in the paper, both scenarios are possible. Due to fiber drawing, the solidification of the silica matrix is accompanied by the “freezing-in” of high-energy structures (3- and 4-MRs) as well as point defect precursors in the final glass. By exposure to radiation, the small-sized rings break up and forms point defects which add to those already present. To clarify this aspect, we reformulated the text between lines 120-127 and added some references.
-
In order to comply with this recommendation, we expanded the conclusions by referencing the newly added section 5.
-
Thanks for the corrections. We modified the text accordingly to the referee’s suggestions.
Round 2
Reviewer 3 Report
I notice a lack of respect on the part of the authors through the answer they gave. The first revision was not in the format requested by the publishing house and this second revision is not either. I am attaching the template that can be found on the magazine platform. THE MARGINS, THE FORMAT OF THE FIGURES, THE CITATION OF THE REFERENCES (the order of the names, the names of the magazines are abbreviated, the quotations of the books with missing information are not respected), the Header and Footer part I consider that a review must be very well documented with many new references. Not old and conference presentations. The graphs shown do not show much, they are common things. There are no comparative tables in which to present variations of the parameters depending on the references. I consider that by copying some sentences from various articles, the authors cannot claim to publish a paper in a Q1 magazine. I recommend to completely revise the article, to complete it with concrete comparative analyzes using several tables and with relevant synthetic graphs, in the format requested by the journal.
